# Epsilon Toxin from *Clostridium perfringens* Induces the Generation of Extracellular Vesicles in HeLa Cells Overexpressing Myelin and Lymphocyte Protein

**DOI:** 10.3390/toxins16120525

**Published:** 2024-12-04

**Authors:** Jonatan Dorca-Arévalo, Antonio Santana-Ruiz, Benjamín Torrejón-Escribano, Mireia Martín-Satué, Juan Blasi

**Affiliations:** 1Department of Pathology and Experimental Therapeutics, Faculty of Medicine and Health Sciences-Campus Bellvitge, University of Barcelona, 08907 Barcelona, Spain; asalcala@gmail.com (A.S.-R.); torrejonbenja@ub.edu (B.T.-E.); martinsatue@ub.edu (M.M.-S.); blasi@ub.edu (J.B.); 2Laboratory of Molecular and Cellular Neurobiology, Bellvitge Biomedical Research Institute (IDIBELL), 08907 L’Hospitalet de Llobregat, Spain; 3Institute of Neuroscience, Bellvitge Health Sciences Campus, University of Barcelona, Carrer de la Feixa Llarga, s/n, 08907 L’Hospitalet de Llobregat, Spain; 4Scientific and Technological Centers (CCiTUB), Bellvitge Campus, University of Barcelona, Carrer de la Feixa Llarga, s/n, 08907 L’Hospitalet de Llobregat, Spain; 5Oncobell Program, Institut d’Investigació Biomèdica de Bellvitge (IDIBELL), CIBERONC, 08908 L’Hospitalet de Llobregat, Spain

**Keywords:** epsilon toxin, *Clostridium perfringens*, myelin and lymphocyte protein, extracellular vesicles

## Abstract

Epsilon toxin (ETX) from *Clostridium perfringens* is a pore-forming toxin (PFT) that crosses the blood–brain barrier and binds to myelin structures. In in vitro assays, ETX causes oligodendrocyte impairment, subsequently leading to demyelination. In fact, ETX has been associated with triggering multiple sclerosis. Myelin and lymphocyte protein (MAL) is widely considered to be the receptor for ETX as its presence is crucial for the effects of ETX on the plasma membrane of host cells that involve pore formation, resulting in cell death. To overcome the pores formed by PFTs, some host cells produce extracellular vesicles (EVs) to reduce the amount of pores inserted into the plasma membrane. The formation of EVs has not been studied for ETX in host cells. Here, we generated a highly sensitive clone from HeLa cells overexpressing the MAL-GFP protein in the plasma membrane. We observed that ETX induces the formation of EVs. Moreover, the MAL protein and ETX oligomers are found in these EVs, which are a very useful tool to decipher and study the mode of action of ETX and characterize the mechanisms involved in the binding of ETX to its receptor.

## 1. Introduction

*Clostridium perfringens*, a pore-forming Gram-positive anaerobic bacterium, is typically found in the gastrointestinal tract of mammals, including humans [1]. The biotyping of *C. perfringens* is performed using letters (A–G) to distinguish different toxin profiles. Among them, the epsilon toxin (ETX) is a pore-forming toxin (PFT), being the third most potent toxin after the tetanus and botulinum neurotoxins. Only types B and D produce the epsilon protoxin (pETX), which exhibits minimal activity but becomes fully active when it undergoes proteolytic cleavage [2].

ETX produces enterotoxemia in ruminants [3,4], causing multiorgan lesions and death. Furthermore, ETX can cross the blood–brain barrier (BBB) [5,6,7], potentially through vascular endothelial cell damage [4] or relaxation of the astrocyte endfeet terminals surrounding blood vessels [7,8,9]. Moreover, ETX produces neuronal damage in murine models [10,11]. Interestingly, ETX has been observed to preferentially accumulate in the kidneys and brain after an acute intravenous injection of ETX in mice, leading to brain edema and the death of epithelial cells in the distal tubules of the kidneys [12,13]. This effect was not observed in mice injected with the non-toxic ETX mutant H106P or pETX [14]. Notably, the Madin-Darby canine kidney (MDCK) cell line, derived from the distal convoluted tubule of dog kidneys, is particularly sensitive to the effects of ETX [15,16,17,18].

Furthermore, various cell lines of different origins show sensitivity to ETX, including Fischer rat thyroid (FRT) epithelial cells [19], MOLT-4 human T lymphoblast cells [20], and mouse lung endothelial 1G11 cells [21]. ETX has also been found to bind to and kill primary human lymphocytes [22] and produce hemolysis in human erythrocytes [23,24], likely through pore formation in the plasma membrane that is linked to the presence of myelin and lymphocyte protein (MAL). MAL is considered to be the ETX receptor in these sensitive cells [25], as recent research has suggested that the MAL protein is a potential receptor for ETX [26].

MAL is an integral membrane protein found in lipid rafts and plays a role in apical protein transport in polarized epithelial cells [27]. MAL is present in the myelin produced by oligodendrocytes and Schwann cells, as well as in the epithelium of the distal collecting duct of the kidneys, the urothelium, and the plasma membrane of T cells, coinciding with the areas where ETX binding is observed [26,28,29,30]. Moreover, overexpression of the MAL protein in different ETX-resistant cell lines lacking endogenous MAL increases their sensitivity to ETX, further supporting its role as an ETX receptor [26]. In addition, it has been reported that the knock-out of MAL gene expression in a T-lymphocyte cell line renders it resistant to ETX [20], while pulldown assays have confirmed a physical interaction between the MAL protein and ETX, providing additional evidence of MAL as the receptor for ETX [20,24].

Curiously, ETX has been associated with the triggering of the demyelinating disease multiple sclerosis (MS). *C. perfringens* types B and D have been detected in the stool samples from MS patients more frequently when compared to healthy control subjects [31,32]. Additionally, antibodies against ETX have been found in the serum of MS patients [32,33]. In vitro experiments have shown that ETX acts on oligodendrocytes [34,35,36], resulting in their death [36] and subsequently causing demyelination [35,36]. However, there is controversy regarding the mechanism of action of ETX in oligodendrocytes, with postulations including pore formation in the plasma membrane [36] or the activation of a pathway mediated by an undefined receptor that does not involve pore formation [35].

It is worth noting that some cells produce microvesicles (MVs) as a defense mechanism against various PFTs, such as streptolysin O, perfringolysin O, and pneumolysin [37,38,39,40], to reduce the presence of pores induced by these toxins in the plasma membrane. In addition, it has been recently published that ETX induces the generation of extracellular vesicles in *Xenopus* oocytes expressing human MAL protein [41]. In the present study, we utilized the HeLa cell line transfected with MAL-GFP to investigate more deeply the mechanism of action of ETX and the cell response. We show that ETX induces the budding of the plasma membrane, inducing extracellular vesicles (EV) formation, which is dependent on MAL. This opens a new field of study to decipher the mode of action of ETX.

## 2. Results

### 2.1. The Generated Clone of MAL-GFP-Expressing HeLa Cells Is Highly Sensitive to ETX

We performed cell subcloning from a population of stably transfected MAL-GFP-expressing HeLa cells [20] to obtain a pure and homogeneous cell population in order to study more accurately the effects of ETX on MAL-GFP-overexpressing HeLa cells and also to avoid the interference from non-MAL-expressing cells.

The selected MAL-GFP-overexpressing HeLa cell clone was incubated with the anti-MAL antibody E-1 to first corroborate MAL protein expression in the plasma membrane. While the MAL protein was not observed in the plasma membrane of the control GFP-overexpressing cells (Figure 1A–C), MAL protein immunostaining was positively detected in the MAL-GFP overexpressing cells (Figure 1D–F).

Additionally, the MTS assays revealed that HeLa cells overexpressing MAL-GFP were highly sensitive to ETX (in red, Figure 2), with a CT_50_ of 1.03 nM (95% CI, 0.87 to 1.22), thereby showing similar CT_50_ values as MDCK cells [21,30]. No cytotoxicity was detected in the GFP-overexpressing HeLa cells (in black, Figure 2) incubated with ETX, confirming previous observations that MAL is crucial for ETX activity (Figure 2).

### 2.2. ETX Induces the Formation of EVs in MAL-GFP-Expressing HeLa Cells

Some cells produce MV in response to some PFTs [37,42,43]. As ETX belongs to the PFT family, we wondered if ETX also induced MV formation in host cells. As there are no studies on ETX and MV formation, we checked this possible effect by incubating MAL-GFP-overexpressing cells with ETX.

MAL-GFP-overexpressing HeLa cells were incubated with pETX-633 or ETX-633 for 40 min, and the effects of ETX on MV formation were analyzed. pETX-633 and ETX-633 bound to the cell plasma membrane at 5 min of incubation (in red, Figure 3, Appendix A). After 20 min of exposure, the binding of pETX-633 on the plasma membrane did not change and no MV were detected (B and E, Figure 3 and Appendix A). Curiously, the binding pattern of ETX-633 on the plasma membrane was different, with a time-dependent decrease in ETX-633 binding (H and K, Figure 3 and Appendix A) and the appearance of MV secreted by the cell (arrows in J and L, Figure 3 and Appendix A).

It can be observed that the ETX action generates MVs of varying sizes, consistent with the size range of EVs reported in the literature [44,45].

### 2.3. Characterization of the EVs Released in Response to ETX

To characterize the production of EVs induced by ETX, the differential ultracentrifugation protocol was performed and the protein composition of the obtained EVs from each centrifugation step was analyzed by Western blot. We isolated the large EVs (LEVs) at 2000 g, the medium EVs (MEVs) at 10,000 g, and the small EVs (SEVs) at 150,000 g. The protein content-based EV characterization was conducted following the minimal information for studies of extracellular vesicles (MISEV) [45,46]. According to the guidelines, CD63, ITGB2, and LAMP1 were used as transmembrane markers, while ANXA2, TSG101, and HSP70 were used as markers of cytosolic proteins recovered from EVs. Calnexin (CANX) and ribosomal proteins (RPL30) were used as purity control markers.

pETX and ETX were detected at all the EV samples as a monomer (black and dark gray arrow, respectively, in Figure 4).

Moreover, ETX was also detected as an oligomer complex only in the MAL-expressing cells (black arrowhead, Figure 4). In addition, the MAL protein was mostly purified from the fraction obtained at 10,000 g, corresponding to MEVs. A faint band corresponding to the MAL protein was detected in the LEVs, while no band was observed in the SEVs. The ETX monomer was not detected in the SEVs obtained from HeLa cells expressing GFP alone, as expected.

pETX did not induce the secretion of any EVs in the MAL-GFP-expressing HeLa cells or in the GFP-expressing HeLa cells. However, ETX induced the formation of LEVs, MEVs, and SEVs only in the MAL-GFP-expressing HeLa cells. Curiously, the biochemical composition of the LEVs, MEVs, and SEVs was quite different, with the MEVs being the most different compared to the LEVs and SEVs. Transmembrane proteins were detected in all EVs, but in lower amounts in the MEVs compared to the LEVs and SEVs. Some of the cytosolic proteins recovered from the MEVs, such as ANXA2 and TSG101, were similar to those recovered from the LEVs and SEVs. CANX was only detected in the LEVs, which are very big and dense. RLP30 were detected only in the LEVs and SEVs, not in the MEVs.

## 3. Discussion

To study cell blebbing and EV formation in response to ETX, we generated a cell model that overexpresses the MAL protein and compared it to non-MAL-expressing cells. We showed that the generated clone expressed human MAL protein in the plasma membrane and that ETX bound to it and induced cell cytotoxicity. Furthermore, the clone presented a CT_50_ value of 1.03 nM (95% CI, 0.87 to 1.22), which is quite similar to that of MDCK cells that show a CT_50_ value of 0.64 nM (95% CI, 0.52 to 0.78) [20,21]. This clone was three times more sensitive to ETX than stably transfected MAL-GFP-expressing HeLa cells, which showed a CT_50_ value of 3.36 nM (95% CI, 2.63 to 4.29) [20]. ETX produced blebbing in MAL-GFP-expressing HeLa cells after 20 min of exposure, and some EVs of different sizes were released from the plasma membrane (arrows, J and L in Figure 3 and Appendix A). To analyze the composition of the EVs, a differential ultracentrifugation protocol using the media from cell cultures incubated with pETX or ETX was performed. While ETX induced the generation of EVs of different sizes, we ruled out the presence of apoptotic bodies since ETX did not elicit caspase-3 cleavage [41].

Some PFTs induce EV formation in order to eliminate the pores inserted into the plasma membrane [37,42,43]. The mechanisms by which these EVs mitigate damage are currently under investigation. It has been proposed that membrane repair occurs in two distinct steps: intrinsic repair and patch formation [37].

Intrinsic repair, which may function in an ATP- and protein-independent manner, refers to the lipid bilayer’s inherent ability to resist PFTs through the biochemical and biophysical properties of membrane lipids. Lipid-modifying and lipid-binding enzymes, particularly sphingomyelinases, likely enhance and regulate this intrinsic repair process. Additionally, calcium influx through pores, combined with intrinsic repair, could initiate an intracellular response and trigger membrane shedding [42]. This calcium influx, promoted by the pore, could activate scramblase activity mediated by TMEM16F/ANO6 channels [41,47,48].

Certain repair proteins, such as Annexins and the ESCRT machinery, may also be recruited to sites of damage [49,50,51,52,53]. These proteins could play a critical role in sealing the damage and facilitating patch repair [54].

In line with these results, in *Xenopus* oocytes, ETX activates calcium-activated chloride channels (CaCCs) via intracellular Ca^2+^ mobilization, inducing the activation of the scramblase and the formation of EVs that carry ETX heptamers [41]. Similarly, we speculate that EV formation in MAL-GFP-overexpressing HeLa cells could involve scramblases, a group of enzymes that facilitate the exchange of phospholipids between the two leaflets of the membrane [55]. However, further studies are needed to validate this hypothesis in mammalian cells.

Early studies on EVs were conducted in cancer research, where EVs were classified into three groups: apoptotic bodies, large extracellular vesicles, and exosomes [45,56]. Since the apoptotic pathway is not involved in the action of ETX [41], we adapted the terminology accordingly: large extracellular vesicles (LEVs), medium extracellular vesicles (MEVs), and small extracellular vesicles (SEVs).

We observed that ETX oligomerizes only if the MAL protein is expressed. An analysis of different types of EVs revealed that the MAL protein was found in some EVs following ETX exposure, with the MEVs (isolated at 10,000 g) being the main EV population in which the MAL protein was present. We propose that ETX binds to the MAL protein and oligomerizes, producing a pore that disrupts the plasma membrane integrity. This induces a rapid loss of intracellular K^+^ ions and an increase in Cl^-^ and Na^+^ levels, subsequently eliciting a Ca^2+^ influx [17] and TMEM16/ANO6 activation as well as the release of EVs containing ETX oligomers and MAL among other proteins. These effects are similar to those of the *Listeria* toxin [43]. Alternative binding partners for ETX, such as the O-glycosylated hepatitis A virus cellular receptor 1 (HAVCR1) protein, caveolin-1 (CAV1), and sulfatide, have been proposed, but they show lower binding efficiency [57,58,59]. These alternative binding partners could explain the presence of ETX oligomers at the SEVs in which the MAL protein was not detected as well as the detection of ETX monomers at the LEVs and MEVs from GFP-expressing HeLa cells that did not express the MAL protein. They could also explain the ability of ETX to oligomerize and form pores in synthetic lipid bilayers in which the MAL protein is not present [17,60], probably due to the presence of a required lipid environment. This occurs at a much lower capacity because ETX pores in planar lipid bilayers are formed by spontaneous self-assembly in the presence of 0.1–0.8 µM ETX [60], whereas MAL-GFP-expressing HeLa cells and MDCK cells show cytotoxicity in response to ETX at a CT_50_ of 1.03 nM and 0.64 nM, respectively. Thus, the effect of ETX is a 100-fold more potent in mammalian cells compared to the synthetic lipid bilayers.

pETX induced the generation of a low amount of LEVs in MAL-GFP-expressing HeLa cells, as very faint bands corresponding to CD63, LAMP1, ANXA2, and TSG101 were detected. This could be explained by the fact that pETX is 1000-fold less toxic than ETX [61] but this toxicity is enough to activate the generation of EVs.

It is known that some PFTs induce the generation of EVs, which are produced by plasma membrane budding [43,62].

The origin of EVs secreted by plasma membrane budding (MEVs) differs from that of exosomes or smaller vesicles (SEVs) [63]. SEVs are enriched in some transmembrane proteins (CD63, ITGB2, and LAMP1) and RNA (RPL30), which aligns with typical exosomal markers originating from multivesicular bodies through an early endosome pathway [46,63].

In contrast, MEVs contain ANXA2 and, to a lesser extent, TSG101 (a component of the ESCRT-I complex). These proteins are believed to be recruited to sites of membrane damage, where they help seal and facilitate membrane repair in response to PFT action [49]. Furthermore, LEVs are the largest EVs generated by the cells and display all the tested markers, similar to the total homogenate, which may correspond to cell fragments, potentially associated with the necrotic pathway induced by ETX [64,65].

Altogether, ETX induces the generation of EVs from host cells, with the majority of the MEVs containing the MAL protein among others. This finding is very useful to understand, characterize, and decipher the mode of action of ETX in the presence or absence of certain proteins that are involved in this process.

## 4. Materials and Methods

### 4.1. Cell Lines

The epithelial cell line from a human cervical adenocarcinoma, HeLa (CCL-2 and ATCC), was selected as it does not express MAL and is not sensitive to ETX.

HeLa cells were grown as explained in [20] and cell cloning from the HeLa cells stably transfected with pEGFPN1-hMAL or pEGFPN1 was performed by applying the limiting dilution assay [66].

### 4.2. Cloning, Expression, Purification, and Activation of pETX and pETX-633

In all cases, wild-type DNA was cloned into the pGEX-4T-1 vector (Amersham Biosciences; Freiburg, Germany) to produce the corresponding recombinant fusion protein, as previously described [11,67,68]. Protein expression protocol was carried out as described in [14,29,30].

Some eluted pETX was labeled with DyLight™ 633 NHS Ester (#46414, Thermo Scientific, Waltham, MA, USA), following the manufacturer’s instructions.

Purified pETX and pETX-633 were incubated with trypsin to form active ETX and ETX-633, respectively [69]. Preparations containing pETX and pETX-633 were incubated with trypsin-agarose beads (Sigma-Aldrich, St. Louis, MO, USA) at room temperature for 60 min, before the removal of the trypsin-coated beads by centrifugation. Protein concentration was determined following the Bradford method [70] using bovine serum albumin (BSA) as the standard.

In total, 10 µL of each sample was electrophoresed in a precast polyacrylamide gel (Mini-PROTEAN^®^ TGX TM; #456-9033, Bio-Rad, Gaithersburg, MD, USA), and Coomassie blue staining was performed to analyze the purity of the recombinant protein.

### 4.3. Cytotoxicity Assays

The cytotoxic effect of ETX was measured using the MTS (3-(4,5- dimethylthiazol-2-yl)-5-(3-carboxymethoxyphenyl)-2-(4-sulfophenyl)-2H tetrazolium) colorimetric assay. Cells were exposed to increasing concentrations of ETX (0, 0.25, 0.5, 1, 2, 5, 10, 25, 50, and 100 nM) for 1 h at 37 °C, and the cytotoxicity assay was carried out as previously described [21]. The results from the cytotoxicity assays were analyzed by nonlinear regression analysis using a two-way ANOVA followed by Šídák’s multiple comparisons test. CT_50_ values were determined as explained previously [21].

### 4.4. Confocal Microscopy

Transfected HeLa cells were grown and fixed with 4% paraformaldehyde, and an immunofluorescence assay was performed as explained previously [20]. MAL protein was stained, incubating the cells with the anti-MAL E1 (Table 1), and was detected using Alexa Fluor 555 goat anti-mouse as a secondary antibody (1:500 dilution, A21424 Thermo Scientific, Waltham, MA, USA). Nuclei were stained with TO-PRO-3 iodide as explained previously [21].

The viability of the transfected HeLa cells was recorded in live-cell imaging time-course experiments, in which the cells were exposed to 25 nM ETX-633 (or pETX-633 for the negative control) for 40 min at 37 °C. The viability protocol was carried out as explained previously [21]. Images were recorded every 77 s for 40 min in each condition. pETX-633 or ETX-633 was added after the first 10 images were recorded as a baseline.

### 4.5. Microvesicle Isolation Protocol

Extracellular vesicles (EVs) were recovered from the different steps of the differential ultracentrifugation (DUC) protocol, which is classically used to isolate small extracellular vesicles or exosomes (SEVs), and the analysis of their protein composition was performed [47,48].

Briefly, cells grown in 10 cm plates were washed twice with PBS and incubated with 50 nM ETX, pETX, or PBS in DMEM-F12 serum-free media for 60 min at 37 °C and 5% CO_2_. The media were collected and centrifuged at 300× *g* for 10 min at 4 °C. The resulting pellets were washed twice with cold PBS and kept on ice, while the supernatants were centrifuged at 2000× *g* for 10 min at 4 °C. The resulting pellets (large extracellular vesicles, LEVs) were washed twice with cold PBS and kept on ice, while the supernatants were centrifuged at 10,000× *g* for 35 min at 4 °C. The resulting pellets (medium extracellular vesicles, MEVs) were washed twice with cold PBS and kept on ice, while the supernatants were centrifuged at 150,000× *g* for 90 min at 4 °C. Finally, the resulting pellets (small extracellular vesicles, SEVs) were washed twice with cold PBS and kept on ice, while the supernatants were discarded. Samples were quantified by the Bradford assay. To characterize the EV composition, 80 μg of each sample supplemented with loading buffer was boiled and subjected to SDS-PAGE and Western blot analysis using the appropriate primary antibodies (Table 1).

## Figures and Tables

**Figure 1 toxins-16-00525-f001:**
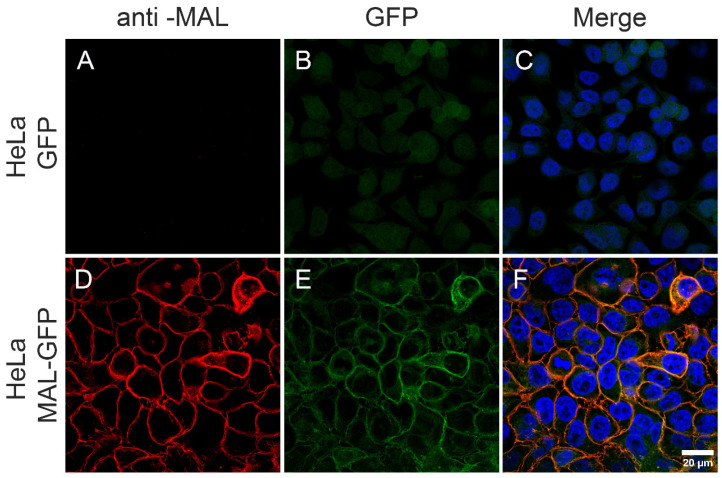
The MAL-GFP-expressing HeLa cell clone expresses high levels of the MAL protein in the plasma membrane. Cells were incubated with an anti-MAL antibody to detect MAL protein expression in the cell plasma membrane. The presence of MAL was compared between the GFP-expressing HeLa cells that did not express MAL (**A**–**C**) and the MAL-GFP-expressing HeLa cells (**D**–**F**). The MAL protein detected by the antibody is shown in red (**A**,**D**), while the MAL protein in fusion with GFP is shown in green (**B**,**E**). (**C**,**F**) The merge in each case. Note the complete colocalization between the antibody binding sites and the MAL-GFP protein distribution (**F**) as well as the absence of MAL expression in the HeLa cells expressing GFP alone (**C**). Nuclei were stained in blue.

**Figure 2 toxins-16-00525-f002:**
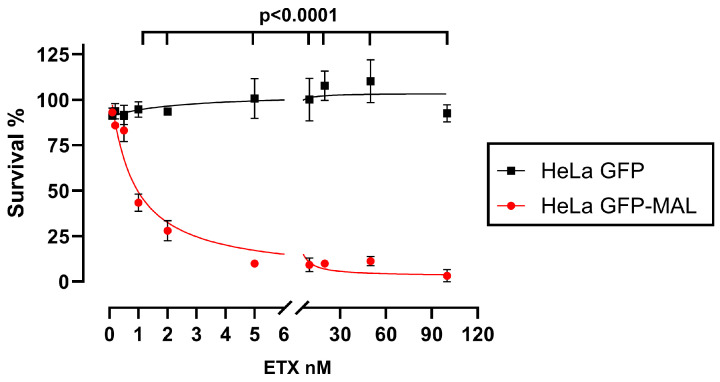
The MAL-GFP-expressing HeLa cell clone is highly sensitive to ETX. Cytotoxicity assays revealed a high sensitivity of the MAL-GFP-expressing HeLa cells (red line) to ETX, while no toxicity was detected in the non-MAL-expressing HeLa cells that expressed GFP alone (black line). Cells were incubated with increasing concentrations of ETX (0, 0.25, 0.5, 1, 2, 5, 10, 25, 50, and 100 nM) for 1 h at 37 °C. Controls were obtained by omitting ETX in each condition (100% cell viability) or by adding 0.2% Triton X-100 (100% cell lethality). Triplicates of the assay were performed in three independent experiments for each condition. Statistical analyses were undertaken by performing a nonlinear regression analysis using a two-way ANOVA followed by Šídák’s multiple comparisons test (*p* < 0.0001).

**Figure 3 toxins-16-00525-f003:**
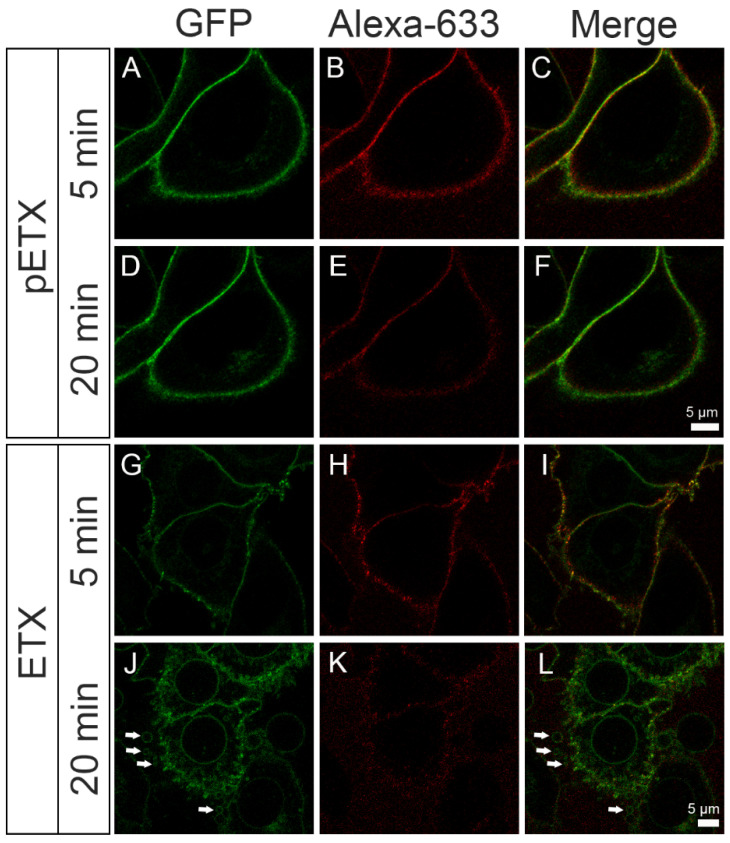
ETX induces the formation of extracellular vesicles in MAL-GFP-expressing HeLa cells. MAL-GFP-expressing HeLa cells (green) were incubated with 25 nM pETX-633 or ETX-633 (red) at 37 °C and under 5% CO_2_ for 40 min. Images were obtained at 5 min and 20 min in each condition. pETX-633 bound to the plasma membrane of the cell, and the binding pattern did not change during the first 20 min of exposure (**A**–**F**). ETX-633 also bound to the plasma membrane (**G**–**L**), but after 20 min of exposure, the amount of bound ETX-633 decreased notably (**K**). In addition, the pattern of MAL-GFP staining changed over the time-course, and extracellular vesicles generated in response to ETX-633 were detected (arrows, **J**,**L**).

**Figure 4 toxins-16-00525-f004:**
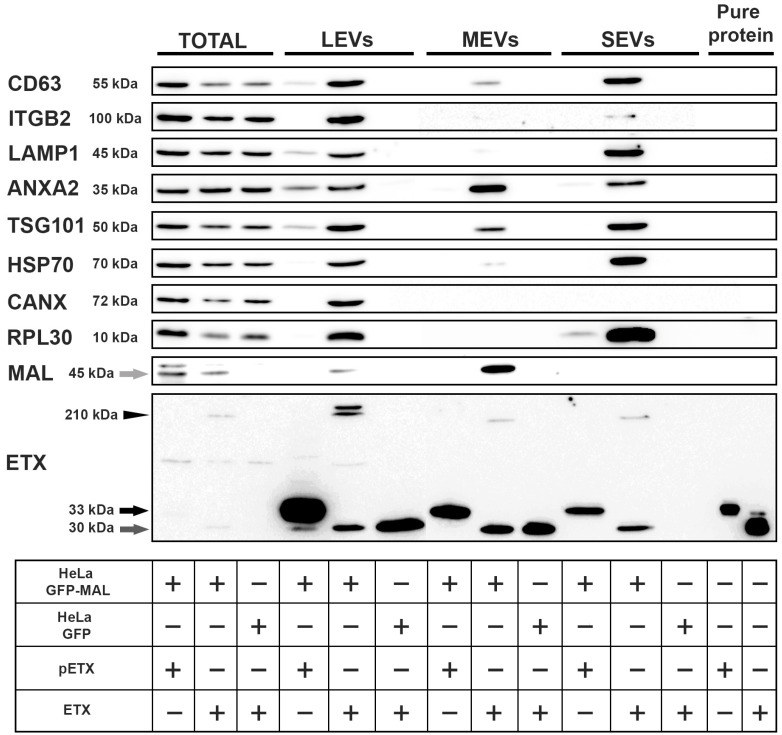
ETX induces the formation of extracellular vesicles with a high content of the MAL protein. Cell media were collected, and a differential ultracentrifugation protocol was applied to isolate the different types of vesicles. ETX oligomer complexes (ETX heptamers) were detected in all the generated vesicles (arrowhead), as well as the monomer forms of both pETX and ETX (black and dark gray arrows, respectively). Moreover, the MAL protein was purified mostly from the MEVs compared to the total sample and the other vesicles (light gray arrow). (Total, total protein sample; LEVs, large extracellular vesicles; MEVs, medium extracellular vesicles; SEVs, small extracellular vesicles). The last two lanes correspond to the pure pETX and ETX proteins, respectively. Samples were run in two gels; total and LEVs were run in one gel, and MEVs, SEVs, and pure protein were run in another gel.

**Table 1 toxins-16-00525-t001:** Table of antibodies used. Table of antibodies used. ANXA: anti-annexin A2; CANX: anti-calnexin; CD63: anti-CD63; ETX: anti-epsilon toxin; HSP70: anti-heat shock protein 70; ITGB2: anti-integrin β2; LAMP1: anti-lysosomal associated membrane protein 1; MAL: anti-myelin and lymphocyte protein; RLP30: anti-receptor like protein 30; TSG101: anti-tumor susceptibility gene 101.

Antibody	Dilution	Host	Precedence
ANXA2	1/1000	Rabbit	ab41803, Abcam
CANX	1/1000	Rabbit	ab133615, Abcam
CD63	1/1000	Rabbit	ab134045, Abcam
ETX	1/1000	Rabbit	In-house
HSP70	1/1000	Rabbit	ab181606, Abcam
ITGB2	1/1000	Mouse	MBS9231826, abYntek
LAMP1	1/1000	Rabbit	ab24170, Abcam
MAL	1/300	Mouse	SC-390687, Santacruz
RLP30	1/1000	Rabbit	C119664, LSBio
TSG101	1/1000	Rabbit	ab125011, Abcam

## Data Availability

The original contributions presented in this study are included in this article and Appendix A. Further inquiries can be directed to the corresponding author.

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
