# Peer review of "Epsilon Toxin from Clostridium perfringens Induces the Generation of Extracellular Vesicles in HeLa Cells Overexpressing Myelin and Lymphocyte Protein"

_toxins, 2024, doi:10.3390/toxins16120525_

Round 1
Reviewer 1 Report
Comments and Suggestions for Authors
The manuscript untitled “Epsilon toxin from Clostridium perfringens induces the generation of extracellular vesicles in HeLa cells overexpressing myelin and lymphocyte protein” uncovers the capacity of ETX to induce EVs in HeLa cells overexpressing its receptor MAL. Through a combination of molecular biology, cellular biology and biochemistry, the authors demonstrate that ETX is able to induce the generation of EVs from host cells. This finding carry substantial implications and may prove valuable to understand, characterize and decipher the mode of action of ETX. The manuscript is clearly written making it accessible to readers who are not familiar with this specific topic and is aligned with the scope of Toxins. However, to further enhance the clarity of the results, I would suggest addressing a few additional points.
Major Points:
· In Figure S1, the authors indicate that the band at approximatively 25kDa corresponds to cleaved caspase 3 but when caspase 3 is cleaved two bands should not appear? one of 19 and another of 17kDa? The authors can explain this difference in the size of the expected band corresponding to cleaved caspase 3? In the original gel of Figure S1, two strong bands appear at the top of the gel, the authors can explain these bands? Unspecific detection?
· In Figure 4, the authors conclude that all EVs are composed by ETX since the authors detected ETX in all the EVs samples by Western Blot. How the authors are sure about the fact that ETX is detected inside the EVs and not outside (bound to extracellular membranes)? Have the authors done, for example, a proteinase K assay with the EVs to confirm that ETX is inside the EVs?
Minor Points:
· General comments: Bacterial species should be in italic (Line 64, 77, 223)
· Line 80: For the first time EV appears in the text it should be written in full
· Line 83: Avoid “newly”
· Line 92-93: The authors mention the panel D-F in the Figure 1 first. Change the order in the Figure 1.
· Figure 1: Eliminate space between “anti” and “-MAL”
· Figure 2: Indicate the p-value (p<0.0001) instead of the stars. In x-axis, write ETX in caps.
· Line 112: Eliminate space between “non-MAL-“ and “expressing”
· Figure S1: To be clearer, markers should be added and the name of the protein should be indicated next to the arrows
· Line 140: Replace “,” by “.”
· Figure 4: To be clearer, markers should be added and the name of the protein should be indicated next to the arrows
· Line 272: The authors mention “mutant DNA”, I don't see a mutant in this work
· Line 280: Eliminate space between “Sepharose” and “4B”
· Line 342: Replace “1 x 106” by “1 x 106”
· Line 381: Replace “were analyzed” by “were quantified”
Author Response
Reviewer 1
The manuscript untitled “Epsilon toxin from Clostridium perfringens induces the generation of extracellular vesicles in HeLa cells overexpressing myelin and lymphocyte protein” uncovers the capacity of ETX to induce EVs in HeLa cells overexpressing its receptor MAL. Through a combination of molecular biology, cellular biology and biochemistry, the authors demonstrate that ETX is able to induce the generation of EVs from host cells. This finding carry substantial implications and may prove valuable to understand, characterize and decipher the mode of action of ETX. The manuscript is clearly written making it accessible to readers who are not familiar with this specific topic and is aligned with the scope of Toxins. However, to further enhance the clarity of the results, I would suggest addressing a few additional points.
Major Points:
- In Figure S1, the authors indicate that the band at approximatively 25kDa corresponds to cleaved caspase 3 but when caspase 3 is cleaved two bands should not appear? one of 19 and another of 17kDa? The authors can explain this difference in the size of the expected band corresponding to cleaved caspase 3? In the original gel of Figure S1, two strong bands appear at the top of the gel, the authors can explain these bands? Unspecific detection?
The authors thank the reviewer for this insightful comment. It is correct that pro-caspase-3 is approximately 34 kDa, while cleaved caspase-3 typically appears as two bands of 19 kDa and 17 kDa, at least in wild-type cells treated with apoptotic inducers. Additionally, there is evidence of high molecular weight bands (>70 kDa) in NIH/3T3 cells corresponding to non-specific antibody binding, as the reviewer noted (https://www.abcam.cn/products/primary-antibodies/cleaved-caspase-3-antibody-epr21032-ab214430.html). Thus, it is possible that the higher molecular weight bands observed in our results could correspond to non-specific antibody binding. Moreover, activated caspase-3 has also been reported to appear as three bands (19, 20, and 24 kDa).
We used the #9661 antibody from Cell Signaling Technology, which has not been reported to produce non-specific bands. However, the molecular weight range in the provided gel image was cropped at 28 kDa, so it is unclear whether higher-weight bands were present.
We assumed that the band around 25 kDa observed in the staurosporine treatment might be associated with the activation of the apoptotic pathway. Similarly, the band around 37 kDa, visible in all samples, could correspond to non-cleaved caspase-3. For this reason, we focused on the band at 25 kDa, assuming it represented cleavage. The absence of a clear 19 kDa band may be related to post-translational modifications in our transfected cells, for that reason we observe the band at 25 kDa.
In any case, to avoid misunderstandings, we have decided to remove Supplementary Figure S1.
- In Figure 4, the authors conclude that all EVs are composed by ETX since the authors detected ETX in all the EVs samples by Western Blot. How the authors are sure about the fact that ETX is detected inside the EVs and not outside (bound to extracellular membranes)? Have the authors done, for example, a proteinase K assay with the EVs to confirm that ETX is inside the EVs?
The authors thank the reviewer for this comment. With the technique used, it is not possible to conclusively determine whether ETX is located inside the EVs or associated with their membrane. We can only confirm the presence of ETX in the EV samples, without specifying its exact localization. Changes have been made in the new re-edited manuscript (lines 153, 236 and 237), replacing "in" with "at." Further studies would be required to confirm whether ETX is inside the EVs or associated with their membrane, as the reviewer suggests.
Minor Points:
- General comments: Bacterial species should be in italic (Line 64, 77, 223)
Changes have been done in the new re-edited manuscript (lines 64, 77 and 212).
- Line 80: For the first time EV appears in the text it should be written in ful
The full name was already included in the abstract (line 12). In any case, the full name has also been included in the text fro the new re-edited manuscript (line 80).
- Line 83: Avoid “newly”
The word has been removed in the new re-edited manuscript (line 84).
- Line 92-93: The authors mention the panel D-F in the Figure 1 first. Change the order in the Figure 1.
The authors thank the reviewer for this comment. In the new re-edited manuscript, the order in the Results section 2.1 (lines 91-93) has been revised to ensure consistency with the figure legend. This adjustment enhances the alignment between the text and the figure.
Figure 1: Eliminate space between “anti” and “-MAL”
Space has been eliminated in the new re-edited manuscript (line 96).
- Figure 2: Indicate the p-value (p<0.0001) instead of the stars. In x-axis, write ETX in caps.
Changes has been done in the new re-edited manuscript.
- Line 112: Eliminate space between “non-MAL-“ and “expressing”
The space has been eliminated in the new re-edited manuscript (line 112).
- Figure S1: To be clearer, markers should be added and the name of the protein should be indicated next to the arrows
Figure S1 has been deleted in the new re-edited manuscript.
- Line 140: Replace “,” by “.”
Changes has been done in the new re-edited manuscript (line 120).
- Figure 4: To be clearer, markers should be added and the name of the protein should be indicated next to the arrows
Changes has been done and a new Figure 4 and the figure legend have been re-edited in the new version of the manuscript.
- Line 272: The authors mention “mutant DNA”, I don't see a mutant in this work
The authors sincerely thank the reviewer for this comment. Changes have been made, and the mutant DNA has been removed from the main text in the new re-edited manuscript (line 275).
Line 280: Eliminate space between “Sepharose” and “4B”
In the new version of the manuscript, the Material and Method section has been re-edited and simplified and Sepharose 4B section has been deleted.
- Line 342: Replace “1 x 106” by “1 x 106”
The section 4.6. caspase 3 activity analysis has been eliminated from the final version of the new re-edited manuscript.
- Line 381: Replace “were analyzed” by “were quantified”
The replace has been done in the new re-edited manuscript (line 330).

Reviewer 2 Report
Comments and Suggestions for Authors
Epsilon toxin (ETX) induces the generation of EV in HeLa cells. Myelin and lymphocyte protein (MAL) is the cell receptor of ETX. To overcome the pores formed by PFTs, some host cells produce extracellular vesicles (EVs) to reduce the amount of pores inserted into the plasma membrane. The formation of EVs has not been studied before for ETX in host cells.
The author showed that ETX induces the formation of EVs, and the MAL protein and ETX oligomers are found in these EVs. These are interesting studies that show the relationship between EV and ETX. I have some questions to confirm.
Figure 1
GFP expression seems different in GFP and MAL-GFP (Fig1B and E).
Please explain this.
C and F correspond to the merge in each case.>I am not sure the meaning. Please explain this. It (C and F) seems the merge between nucleation stained and MAL.
FigureS1 The bands do not show the same position between pETX and pETXpure.
In Figure 4, there are the same difference between pETX and pETXpure.
Please explain the difference.
Figure 3 ETX induces the formation of extracellular vesicles in MAL -GFP -expressing HeLa cells.
The appearance of EVs secreted by the cell (arrows in J and L, Figure 3 and Video S2).
The author said the arrows show the EVs, but please explain how the author could identify these as EV.
p7 L188
Does oligomer complex mean the heptamer pore?
The author purified the LEVs, large extracelluar vesicles; MEVs, medium extracellular vesicles; SEVs, small extracellular vesicles by the centrifugation. To classify into three groups are the general way in EV stuides?
I am not sure the EV's characters.
Was there any knowledge (references) that there are differences among EVs' sizes? Please explain the differences among the three EVs with references.
As the author said, "To overcome the pores formed by PFTs, some host cells produce extracellular vesicles (EVs) to reduce the amount of pores inserted into the plasma membrane.". Please explain the meaning of how EV can overcome the pores formed by PFTs.
p7 L194
pETX did not induce the secretion of any EVs in the MAL-GFP-expressing HeLa cells or in the GFP-expressing HeLa cells. However, ETX induced the formation of LEVs, MEVs and SEVs only in the MAL-GFP-expressing HeLa cells.
In another pore-forming toxin, have similar facts been known before?
p7 L218
we ruled out the presence of apoptotic bodies since ETX did not elicit caspase-3 cleavage.
What is the difference between EV and apoptotic bodies?
p7 L196
Curiously, the biochemical composition of the LEVs, MEVs, and SEVs was quite different, with the MEVs being the most different compared to the LEVs and SEVs.
This is strange to me. Is there any explanation for the fact that the MEV has a specific feature?
Figure S1
I understand ETX causes cell death, not by the apoptotic pathway. Could the author check the pyroptosis pathway? Please add comments on this. (C3 exoenzyme triggers the pyroptosis pathway.)
Author Response
Reviewer 2
Epsilon toxin (ETX) induces the generation of EV in HeLa cells. Myelin and lymphocyte protein (MAL) is the cell receptor of ETX. To overcome the pores formed by PFTs, some host cells produce extracellular vesicles (EVs) to reduce the amount of pores inserted into the plasma membrane. The formation of EVs has not been studied before for ETX in host cells.
The author showed that ETX induces the formation of EVs, and the MAL protein and ETX oligomers are found in these EVs. These are interesting studies that show the relationship between EV and ETX. I have some questions to confirm.
Figure 1
GFP expression seems different in GFP and MAL-GFP (Fig1B and E).
Please explain this.
We thank the reviewer for this comment. The difference in GFP expression observed in Figures 1B and 1E is attributed to the overexpression of the MAL-GFP protein. It is important to note that MAL is a transmembrane protein localized to the plasma membrane, whereas GFP alone is found in the cytosol and not in the plasma membrane. Consequently, the distribution patterns differ: MAL-GFP is predominantly localized to the plasma membrane, while GFP alone remains confined to the cytosol, resulting in a more diffuse staining pattern.
C and F correspond to the merge in each case.>I am not sure the meaning. Please explain this. It (C and F) seems the merge between nucleation stained and MAL.
We thank the reviewer for this comment. The term "merge" refers to the superposition of all images in each case, combined with the nuclei staining. For example, the merge in panel C represents the superposition of panels A, B, and the nuclei. Similarly, the merge in panel F corresponds to the superposition of panels D, E, and the nuclei.
FigureS1 The bands do not show the same position between pETX and pETXpure.
The Figure S1 has been eliminated in the final version
In Figure 4, there are the same difference between pETX and pETXpure.
Please explain the difference.
We thank the reviewer for this comment. Based on our observations, there is no difference in the band positions between pETX and pETX pure. In the annexed figure 4, we have added a red line to indicate that all bands have the same molecular weight.
Figure 3 ETX induces the formation of extracellular vesicles in MAL -GFP -expressing HeLa cells.
The appearance of EVs secreted by the cell (arrows in J and L, Figure 3 and Video S2).
The author said the arrows show the EVs, but please explain how the author could identify these as EV.
We thank the reviewer for this comment. Indeed, this is a key question that motivated us to investigate whether the observed vesicles could correspond to extracellular vesicles (EVs) generated by ETX action. These vesicles appeared to form and be released from the plasma membrane. Based on their size, we hypothesized that they were EVs. Specifically, we have observed particles of different sizes (approximately of 2 µm, 1 µm, and 500 nm) as far as it can be resolved by confocal microscopy, which align with the size ranges for EVs reported in the literature [1,2]. Additionally, our results demonstrated that MAL-GFP was purified in the LEVs fraction, corresponding to particles ranging from 100 nm to 1 µm [1,2].
Furthermore, it is known that some pore-forming toxins induce the generation of EVs [3,4]. For these reasons, we hypothesized that the particles observed could correspond to EVs. A new paragraph with references has been added to the manuscript to clarify this point in the new re-edited manuscript (lines 132-133).
p7 L188
Does oligomer complex mean the heptamer pore?
We thank the reviewer for this comment. Yes, in this context, 'oligomer' refers to a heptamer pore. This has been incorporated into the endfoot in the new re-edited Figure 4 (line 158).
The author purified the LEVs, large extracelluar vesicles; MEVs, medium extracellular vesicles; SEVs, small extracellular vesicles by the centrifugation. To classify into three groups are the general way in EV stuides?
We thank the reviewer for this comment. The field of microvesicles is constantly evolving, and new nomenclature and classifications are being accepted. In fact, there are more than ten different types of microvesicles, some of which are more widely accepted than others based on factors such as size, composition, origin, biogenesis, and function [5]. Traditionally, extracellular vesicles (EVs) were classified into three groups [2,6]. For this reason (and for simplicity), we have used this classification in our study. Early research on EVs, particularly in cancer, employed a slightly different nomenclature. Large extracellular vesicles (LEVs), larger than 1 µm, were referred to as apoptotic bodies. Vesicles ranging from 100 nm to 1 µm were termed large extracellular vesicles (now known as medium extracellular vesicles, or MEVs), while exosomes, smaller than 100 nm, were classified as small extracellular vesicles (SEVs).
We chose to revise this nomenclature because the generation of these vesicles in our study is not related to the apoptosis pathway. In cancer research, the focus was often on the exosome fraction, with larger vesicles being dismissed as fragments of broken cells or apoptotic bodies. However, since the apoptosis pathway is not activated by ETX action [7], we opted to study all isolated vesicle fractions in each case and adapted the terminology accordingly. A clarified version of this explanation has been added to the discussion section in the new re-edited manuscript (lines 219-223).
I am not sure the EV's characters.
Was there any knowledge (references) that there are differences among EVs' sizes? Please explain the differences among the three EVs with references.
Regarding the previous comment, the references have been added to the explanation above. Additionally, references have been included in the manuscript (lines 219-223).
As the author said, "To overcome the pores formed by PFTs, some host cells produce extracellular vesicles (EVs) to reduce the amount of pores inserted into the plasma membrane.". Please explain the meaning of how EV can overcome the pores formed by PFTs.
The authors thank the reviewer for their insightful comment. We propose that there is a clearance of ETX complexes through sequestration into blebs similarly as it has been also proposed for Streptolysin O, an other pore-forming toxin (PFT) [8]. Moreover, it has been proposed that membrane repair occurs in two distinct steps: intrinsic repair and patch formation [3]. Intrinsic repair, which may function in an ATP- and protein-independent manner, refers to the lipid bilayer's ability to resist PFTs through the biochemical and biophysical properties of membrane lipids, such as sterol accessibility or the sequestration of toxin oligomers onto blebs. Lipid-modifying and lipid-binding enzymes, particularly sphingomyelinases, likely enhance and regulate this intrinsic repair process.
Additionally, calcium influx through pores, in conjunction with intrinsic repair, could initiate an intracellular response and trigger membrane shedding [8], possibly by activating scramblase activity mediated by TMEM16F channels [7]. Certain repair proteins, including Annexins and the ESCRT machinery, may also be recruited to the sites of damage [9-12]. These proteins could play a role in sealing the damage and facilitating patch repair [13]. A new paragraph as been included in the discussion section in the new re-edited manuscript (lines 197-208).
p7 L194
pETX did not induce the secretion of any EVs in the MAL-GFP-expressing HeLa cells or in the GFP-expressing HeLa cells. However, ETX induced the formation of LEVs, MEVs and SEVs only in the MAL-GFP-expressing HeLa cells.
In another pore-forming toxin, have similar facts been known before?
It has been reported that some PFTs, such as streptolysin O and perfringolysin O, can induce the formation of extracellular vesicles [3,4,14], as mentioned previously. More detailed studies have been conducted with the PFT pneumolysin from Streptococcus pneumoniae, where the generation of extracellular vesicles was detected in a T lymphoid cell line as well as in several myeloid cell lines [15]. However, these studies primarily relied on imaging techniques rather than western blotting assays, and they did not analyze extracellular vesicles of different sizes.
In our paper, we analyze for the first time, using both imaging techniques and western blotting assays, the generation of extracellular vesicles of different sizes and compositions induced by ETX, a PFT.
A new reference to pneumolysin has been added to the introduction of the new re-edited manuscript (lines 74-75) to address the reviewer's comment.
p7 L218
we ruled out the presence of apoptotic bodies since ETX did not elicit caspase-3 cleavage.
What is the difference between EV and apoptotic bodies?
The authors thank the reviewer for this insightful comment. Both apoptotic bodies and extracellular vesicles (EVs) are particles released from cells; however, they exhibit distinct characteristics. Apoptotic bodies are larger (>1 µm) and contain fragments of DNA and cytoplasm. They are generated during the activation of the apoptosis pathway, primarily serving to eliminate dead cells without triggering inflammation.
In contrast, EVs, which include both small and large vesicles, are smaller (ranging from 30 nm to 1 µm). They contain a diverse array of proteins, lipids, and RNA and are released via exocytosis or plasma membrane blebbing. The primary roles of EVs involve cell communication and modulation of the immune response [16].
p7 L196
Curiously, the biochemical composition of the LEVs, MEVs, and SEVs was quite different, with the MEVs being the most different compared to the LEVs and SEVs.
This is strange to me. Is there any explanation for the fact that the MEV has a specific feature?
The authors thank the reviewer for this comment. This topic is currently under investigation. It is known that some pore-forming toxins (PFTs) induce the generation of extracellular vesicles (EVs), historically referred to as "ectosomes," which are produced by plasma membrane budding [17,18].
We have seen that MEVs produced by ETX contain ANXA2 and, to a lesser extent, TSG101 (a component of the ESCRT-I complex). These proteins are believed to be recruited to sites of membrane damage, where they help seal and facilitate membrane repair in response to PFT action [3,9]. To our knowledge, the origin of EVs secreted via plasma membrane budding (MEVs) differs from that of exosomes or smaller vesicles (SEVs) [19]. SEVs are enriched in transmembrane proteins (CD63, ITGB2, and LAMP1) and RNA (RPL30), which aligns with typical exosomal markers that originate from multivesicular bodies through the early endosome pathway [2,19].
Furthermore, large extracellular vesicles (LEVs), the largest EVs generated by the cells, display all the tested markers, similar to the total homogenate. These may correspond to cell fragments, potentially associated with the necrotic pathway induced by ETX [20,21]. A new paragraph has been added to the discussion section in the new re-edited manuscript (lines 252-262).
Figure S1
I understand ETX causes cell death, not by the apoptotic pathway. Could the author check the pyroptosis pathway? Please add comments on this. (C3 exoenzyme triggers the pyroptosis pathway.)
The authors thank the reviewer for this comment. As a result, Figure S1 has been removed from the final version of the manuscript.
While some studies suggest that ETX induces necrotic effects in certain epithelial cells [20,21], exploring other cell death pathways, such as pyroptosis, would be interesting, as suggested by the reviewer. Additionally, investigating programmed cell death mechanisms like necroptosis and ferroptosis, as well as non-programmed cell death processes like paraptosis, would also be valuable. However, new and more extensive experiments are needed to achieve these objectives, making this an important area for future research.
- Salem, I.; Naranjo, N.M.; Singh, A.; DeRita, R.; Krishn, S.R.; Sirman, L.S.; Quaglia, F.; Duffy, A.; Bowler, N.; Sayeed, A.; et al. Methods for extracellular vesicle isolation from cancer cells. Cancer Drug Resist 2020, 3, 371-384, doi:10.20517/cdr.2019.118.
- Théry, C.; Witwer, K.W.; Aikawa, E.; Alcaraz, M.J.; Anderson, J.D.; Andriantsitohaina, R.; Antoniou, A.; Arab, T.; Archer, F.; Atkin-Smith, G.K.; et al. Minimal information for studies of extracellular vesicles 2018 (MISEV2018): a position statement of the International Society for Extracellular Vesicles and update of the MISEV2014 guidelines. J Extracell Vesicles 2018, 7, 1535750, doi:10.1080/20013078.2018.1535750.
- Romero, M.; Keyel, M.; Shi, G.; Bhattacharjee, P.; Roth, R.; Heuser, J.E.; Keyel, P.A. Intrinsic repair protects cells from pore-forming toxins by microvesicle shedding. Cell Death Differ 2017, 24, 798-808, doi:10.1038/cdd.2017.11.
- Köffel, R.; Wolfmeier, H.; Larpin, Y.; Besançon, H.; Schoenauer, R.; Babiychuk, V.S.; Drücker, P.; Pabst, T.; Mitchell, T.J.; Babiychuk, E.B.; et al. Host-Derived Microvesicles Carrying Bacterial Pore-Forming Toxins Deliver Signals to Macrophages: A Novel Mechanism of Shaping Immune Responses. Front Immunol 2018, 9, 1688, doi:10.3389/fimmu.2018.01688.
- Welsh, J.A.; Goberdhan, D.C.I.; O'Driscoll, L.; Buzas, E.I.; Blenkiron, C.; Bussolati, B.; Cai, H.; Di Vizio, D.; Driedonks, T.A.P.; Erdbrügger, U.; et al. Minimal information for studies of extracellular vesicles (MISEV2023): From basic to advanced approaches. J Extracell Vesicles 2024, 13, e12404, doi:10.1002/jev2.12404.
- Yáñez-Mó, M.; Siljander, P.R.; Andreu, Z.; Zavec, A.B.; Borràs, F.E.; Buzas, E.I.; Buzas, K.; Casal, E.; Cappello, F.; Carvalho, J.; et al. Biological properties of extracellular vesicles and their physiological functions. J Extracell Vesicles 2015, 4, 27066, doi:10.3402/jev.v4.27066.
- Cases, M.; Dorca-Arévalo, J.; Blanch, M.; Rodil, S.; Terni, B.; Martín-Satué, M.; Llobet, A.; Blasi, J.; Solsona, C. The epsilon toxin from Clostridium perfringens stimulates calcium-activated chloride channels, generating extracellular vesicles in Xenopus oocytes. Pharmacol Res Perspect 2024, 12, e70005, doi:10.1002/prp2.70005.
- Keyel, P.A.; Loultcheva, L.; Roth, R.; Salter, R.D.; Watkins, S.C.; Yokoyama, W.M.; Heuser, J.E. Streptolysin O clearance through sequestration into blebs that bud passively from the plasma membrane. J Cell Sci 2011, 124, 2414-2423, doi:10.1242/jcs.076182.
- Cooper, S.T.; McNeil, P.L. Membrane Repair: Mechanisms and Pathophysiology. Physiol Rev 2015, 95, 1205-1240, doi:10.1152/physrev.00037.2014.
- Demonbreun, A.R.; Quattrocelli, M.; Barefield, D.Y.; Allen, M.V.; Swanson, K.E.; McNally, E.M. An actin-dependent annexin complex mediates plasma membrane repair in muscle. J Cell Biol 2016, 213, 705-718, doi:10.1083/jcb.201512022.
- Babiychuk, E.B.; Monastyrskaya, K.; Potez, S.; Draeger, A. Blebbing confers resistance against cell lysis. Cell Death Differ 2011, 18, 80-89, doi:10.1038/cdd.2010.81.
- Jimenez, A.; Perez, F. Plasma membrane repair: the adaptable cell life-insurance. Current Opinion in Cell Biology 2017, 47, 99-107, doi:10.1016/j.ceb.2017.03.011.
- McNeil, A.K.; Rescher, U.; Gerke, V.; McNeil, P.L. Requirement for annexin A1 in plasma membrane repair. J Biol Chem 2006, 281, 35202-35207, doi:10.1074/jbc.M606406200.
- Brito, C.; Cabanes, D.; Sarmento Mesquita, F.; Sousa, S. Mechanisms protecting host cells against bacterial pore-forming toxins. Cell Mol Life Sci 2019, 76, 1319-1339, doi:10.1007/s00018-018-2992-8.
- Larpin, Y.; Besançon, H.; Iacovache, M.I.; Babiychuk, V.S.; Babiychuk, E.B.; Zuber, B.; Draeger, A.; Köffel, R. Bacterial pore-forming toxin pneumolysin: Cell membrane structure and microvesicle shedding capacity determines differential survival of cell types. FASEB J 2020, 34, 1665-1678, doi:10.1096/fj.201901737RR.
- Akers, J.C.; Gonda, D.; Kim, R.; Carter, B.S.; Chen, C.C. Biogenesis of extracellular vesicles (EV): exosomes, microvesicles, retrovirus-like vesicles, and apoptotic bodies. J Neurooncol 2013, 113, 1-11, doi:10.1007/s11060-013-1084-8.
- Colombo, M.; Raposo, G.; Théry, C. Biogenesis, secretion, and intercellular interactions of exosomes and other extracellular vesicles. Annu Rev Cell Dev Biol 2014, 30, 255-289, doi:10.1146/annurev-cellbio-101512-122326.
- Wu, N.; Cernysiov, V.; Davidson, D.; Song, H.; Tang, J.; Luo, S.; Lu, Y.; Qian, J.; Gyurova, I.E.; Waggoner, S.N.; et al. Critical Role of Lipid Scramblase TMEM16F in Phosphatidylserine Exposure and Repair of Plasma Membrane after Pore Formation. Cell Rep 2020, 30, 1129-1140.e1125, doi:10.1016/j.celrep.2019.12.066.
- Doyle, L.M.; Wang, M.Z. Overview of Extracellular Vesicles, Their Origin, Composition, Purpose, and Methods for Exosome Isolation and Analysis. Cells 2019, 8, doi:10.3390/cells8070727.
- Geng, Z.; Kang, L.; Huang, J.; Gao, S.; Wang, J.; Yuan, Y.; Li, Y.; Xin, W. Epsilon toxin from Clostridium perfringens induces toxic effects on skin tissues and HaCaT and human epidermal keratinocytes. Toxicon 2021, 198, 102-110, doi:10.1016/j.toxicon.2021.05.002.
- Chehreara, R.; Karizi, S.Z.; Hosseini, H.M.; Mirhosseini, S.A.; Shafiei, M.; Amani, J.; Kazemi, R. Cell death analysis of recombinant mature epsilon toxin on the kidney cell line. Iran J Microbiol 2021, 13, 832-838, doi:10.18502/ijm.v13i6.8088.

Round 2
Reviewer 2 Report
Comments and Suggestions for Authors
All of my concerns were addressed adequately.